# First-Trimester Preeclampsia-Induced Disturbance in Maternal Blood Serum Proteome: A Pilot Study

**DOI:** 10.3390/ijms251910653

**Published:** 2024-10-03

**Authors:** Natalia Starodubtseva, Alisa Tokareva, Alexey Kononikhin, Alexander Brzhozovskiy, Anna Bugrova, Evgenii Kukaev, Kamilla Muminova, Alina Nakhabina, Vladimir E. Frankevich, Evgeny Nikolaev, Gennady Sukhikh

**Affiliations:** 1V.I. Kulakov National Medical Research Center for Obstetrics Gynecology and Perinatology, Ministry of Healthcare of Russian Federation, 117997 Moscow, Russia; alisa.tokareva@phystech.edu (A.T.); a_kononihin@oparina4.ru (A.K.); agb.imbp@gmail.com (A.B.); a_bugrova@oparina4.ru (A.B.); e_kukaev@oparina4.ru (E.K.); k_muminova@oparina4.ru (K.M.); a_nahabina@oparina4.ru (A.N.); v_frankevich@oparina4.ru (V.E.F.); g_sukhikh@oparina4.ru (G.S.); 2Moscow Center for Advanced Studies, 123592 Moscow, Russia; 3Emanuel Institute of Biochemical Physics, Russian Academy of Sciences, 119334 Moscow, Russia; 4V.L. Talrose Institute for Energy Problems of Chemical Physics, N.N. Semenov Federal Research Center of Chemical Physics, 119334 Moscow, Russia; 5Laboratory of Translational Medicine, Siberian State Medical University, 634050 Tomsk, Russia; 6Independent Researcher, 121205 Moscow, Russia; ennikolaev@gmail.com

**Keywords:** preeclampsia, prognosis, quantitative proteomics, screening, mass spectrometry, serum, pregnancy

## Abstract

Preeclampsia (PE) is a complex and multifaceted obstetric syndrome characterized by several distinct molecular subtypes. It complicates up to 5% of pregnancies and significantly contributes to maternal and newborn morbidity, thereby diminishing the long-term quality of life for affected women. Due to the widespread dissatisfaction with the effectiveness of existing approaches for assessing PE risk, there is a pressing need for ongoing research to identify newer, more accurate predictors. This study aimed to investigate early changes in the maternal serum proteome and associated signaling pathways. The levels of 125 maternal serum proteins at 11–13 weeks of gestation were quantified using liquid chromatography–multiple reaction monitoring mass spectrometry (LC-MRM MS) with the BAK-125 kit. Ten serum proteins emerged as potential early markers for PE: Apolipoprotein M (APOM), Complement C1q subcomponent subunit B (C1QB), Lysozyme (LYZ), Prothrombin (F2), Albumin (ALB), Zinc-alpha-2-glycoprotein (AZGP1), Tenascin-X (TNXB), Alpha-1-antitrypsin (SERPINA1), Attractin (ATRN), and Apolipoprotein A-IV (APOA4). Notably, nine of these proteins have previously been associated with PE in prior research, underscoring the consistency and reliability of our findings. These proteins play key roles in critical molecular processes, including complement and coagulation cascades, platelet activation, and insulin-like growth factor pathways. To improve the early prediction of PE, a highly effective Support Vector Machine (SVM) model was developed, analyzing 19 maternal serum proteins from the first trimester. This model achieved an area under the curve (AUC) of 0.91, with 87% sensitivity and 95% specificity, and a hazard ratio (HR) of 13.5 (4.6–40.8) with *p* < 0.001. These findings demonstrate that serum protein-based SVM models possess significantly higher predictive power compared to the routine first-trimester screening test, highlighting their superior utility in the early detection and risk stratification of PE.

## 1. Introduction

PE is a multisystem disorder unique to pregnancy that usually appears after 20 weeks of gestation. It is characterized by newly developed hypertension and proteinuria, and it can affect multiple organs including the liver, kidneys, brain, and blood clotting mechanisms [1,2]. This condition impacts mortality rates globally and diminishes the long-term quality of life for affected women [3]. Infants of mothers with preeclampsia are at approximately a two-fold higher risk of neonatal death11 and increased risk of neonatal morbidity, including low Apgar scores, seizures, neonatal encephalopathy, and neonatal intensive care admission [4,5].

Up until recently, the prediction of PE was carried out solely on the basis of confirmed risk factors [6,7]. Due to the low effectiveness of this approach, the Fetal Medicine Foundation (FMF) developed a comprehensive assessment of PE and intrauterine growth restriction (IUGR) based on first-trimester screening of pregnancy [8]. This comprehensive assessment includes biochemical indicators, such as placental growth factor (PlGF) and pregnancy-associated plasma protein A (PAPP-A), biophysical markers, such as mean arterial pressure (MAP) and the uterine artery pulsatility index (UtA-PI), and maternal history. Together, these factors are highly effective predictors of PE, with a detection rate (DR) of 90% for eo-PE, 75% for preterm PE, and 42% for term PE, with a false positive rate (FPR) of 10% [6,7,9,10]. As shown, the effectiveness of predicting late-onset PE is notably limited. Furthermore, despite the development of multiparametric screening, it is not applicable to first-time pregnant women, as we cannot assess the obstetric risk factors in this population. This situation underscores the pressing need for ongoing research to identify newer, more accurate predictors. Multiple reaction monitoring (MRM) is recognized as an advanced and highly refined technique in mass spectrometry (MS)-based proteomics, playing a crucial role in the search for biomarkers [11]. Particularly, the BAK 125 kit developed by MRM Proteomics Inc. (Montreal, QC, Canada) exemplifies this sophisticated technology. This kit, rigorously validated in line with the standards set by the Clinical Proteomic Tumor Analysis Consortium (CPTAC), facilitates the quantification of over a hundred proteins from blood samples, requiring only 10 µL of the sample for analysis. The BAK 125 kit covers an extensive array of potential disease biomarkers, making it a versatile tool in proteomics research. Among these quantified proteins, 44 have received approval from the FDA as biomarkers, underscoring their clinical relevance and utility. Additionally, 51 proteins within this kit are identified as potential biomarkers for cardiovascular diseases [11,12,13].

In this study, we undertook the quantitative analysis of 125 maternal plasma proteins using liquid chromatography–multiple reaction monitoring mass spectrometry (LC-MRM-MS). We aimed to identify specific protein biomarkers and altered signaling pathways, thereby advancing our understanding of PE etiology and potentially informing the development of novel diagnostic and therapeutic strategies.

## 2. Results

### 2.1. Subject Demographics

This study included a cohort of 50 pregnant women, categorized into different groups: preeclampsia (PE), which was subdivided into early-onset PE (eo-PE, *n* = 8) and late-onset PE (lo-PE, *n* = 22), gestational arterial hypertension (GAH, *n* = 7), and a control group (CTR, *n* = 13). Blood serum samples were collected at the first pregnancy screening between 11 and 13 weeks of gestation. Table 1 provides a summary of the demographic and clinical data, while detailed information can be found in Appendix A.

The groups were well matched in terms of maternal age, nicotine use, and histories of somatic and gynecological disorders. On average, 66% of the participants were nulliparous (*p* > 0.05). Notably, the risk of developing PE, as calculated during the first-trimester screening, was similar across all groups (*p* = 0.76). This highlights the critical need for new biochemical markers that can be identified during the first trimester, which would allow for early prediction of placenta-associated complications, including PE, and thereby support timely interventions to improve maternal and fetal outcomes.

A significant difference was observed in the maximal diastolic blood pressure, which was higher in the eo-PE group compared to the lo-PE group (*p* = 0.04). However, there were no statistically significant differences between the PE groups regarding proteinuria levels and platelet counts (*p* > 0.05). Importantly, the PE diagnostic marker sFlt-1/PlGF ratio was significantly elevated in both eo-PE and lo-PE patients compared to the GAH and CTR groups (*p* < 0.005). Additionally, the frequency of intrauterine growth restriction (IUGR) was significantly higher in the eo-PE group compared to the lo-PE group (*p* = 0.002).

All patients diagnosed with early-onset PE delivered prematurely, with a median gestational age of 31.3 weeks. These early deliveries were necessitated due to the deterioration of maternal or fetal health, or both. Furthermore, the frequency of emergency cesarean sections was highest in the early PE group (100%). The greatest challenges in newborn care were also observed in the eo-PE group.

### 2.2. Maternal Serum Quantitative Proteomics (LC-MRM-MS)

A comprehensive quantitative assessment of 125 proteins present in the blood serum of pregnant women was conducted using LC-MRM-MS, employing internal stable isotope-labeled peptide standards. The proteins analyzed account for 99% of the total serum protein mass.

A total of 69 proteins were identified in at least 50% of the serum samples, with their concentrations exceeding the lowest limit of quantification (LLOQ), as detailed in Appendix A. The dynamic range of these proteins spans 4.7 orders of magnitude, as shown in Appendix A. Serum albumin (ALB) and Apolipoprotein A-I (APOA1) were the most abundant proteins in the serum, with concentrations of 278,000 (256,000–286,000) fmol/µL and 49,000 (45,400–55,600) fmol/µL, respectively. Conversely, Cholinesterase and Adipocyte plasma membrane-associated proteins were among the least abundant proteins, with concentrations of 5.31 (4.71–6.06) fmol/µL and 5.62 (2.68–6.74) fmol/µL, respectively. ALB exhibited the least variability, with a coefficient of variation (CV) of only 8%. In contrast, pregnancy zone protein (PZB) showed the highest variation in serum concentration among the studied patients, with a CV of 91% (Appendix A).

A total of seven serum proteins showed a significant increase during early gestation (first trimester) in PE cases, as detailed in Appendix A and depicted in Figure 1. These proteins include Apolipoprotein M (APOM), Complement C1q subcomponent subunit B (C1QB), Lysozyme C (LYZ), Prothrombin (F2), ALB, Zinc-alpha-2-glycoprotein (AZGP1), and Tenascin-X (TNXB). These specific proteins were effective in distinguishing serum samples from PE patients compared to the control and gestational age-matched healthy (GAH) groups, as illustrated in Figure 2A. ALB showed the largest effect size, and another six proteins demonstrated a medium effect size, as measured by Cohen’s d values (Figure 2B). This indicates that ALB had the most substantial differential expression, making it a potentially significant biomarker for early detection of PE.

Furthermore, a significant correlation was observed among seven potential markers for PE, as illustrated in Figure 2C. The strongest correlation was identified between Prothrombin (F2) and Zinc-alpha-2-glycoprotein (AZGP1), with a correlation coefficient of r = 0.44 (*p* < 0.05). Gene Ontology (GO) pathway analysis uncovered a diverse range of biological processes influenced by PE. Notably, the analysis highlighted key networks involving the transcription factors, FOXA2 and FOXA3, as sourced from the PID database. Additionally, alterations in metabolic pathways related to vitamin B12, selenium, and folate were identified, along with significant impacts on complement and coagulation cascades, as detailed in the Wikipathways database. Moreover, the analysis pointed to the regulation of insulin-like growth factor (IGF) transport and uptake, mediated by insulin-like growth factor binding proteins (IGFBPs). Other processes affected include platelet activation, signaling pathways, and aggregation events, as outlined in the Reactome database. These findings point to the multifaceted nature of PE’s impact on maternal serum biochemistry and underscore the relevance of these pathways in understanding the condition. Refer to Appendix A for more details.

A more detailed analysis of the PE group has revealed distinctive features in the maternal serum proteome during the first trimester for two clinical types: eo-PE (≤34 weeks of gestation) and lo-PE (>34 weeks of gestation). In cases of eo-PE, the concentrations of Alpha-1-antitrypsin (SERPINA1) and Attractin (ATRN) were significantly decreased (*p* < 0.05) compared to those observed in lo-PE. Conversely, levels of Apolipoprotein A-IV (APOA4) showed a significant increase (*p* < 0.05) in maternal serum during the early gestation period (Figure 3, Appendix A).

As a result, ten serum proteins have emerged as potential early markers for PE: APOM, C1QB, LYZ, F2, ALB, AZGP1, TNXB, SERPINA1, ATRN, and APOA4. These proteins are involved in common molecular processes, forming a cohesive molecular network, which is depicted in Figure 3C.

Several proteins included in the study panel exhibited statistically significant correlations (*p* < 0.05) with various demographic and clinical parameters of the patients, indicating notable relationships between these biomarkers and the characteristics of the subjects analyzed (Appendix A). Carbonic anhydrase 1 (CA1) and Albumin (ALB) demonstrated a medium inverse association with patient age, with correlation coefficients of r = −0.43 (*p* = 0.002) and r = −0.40 (*p* = 0.004), respectively (Appendix A). Additionally, a number of serum proteins were found to correlate with maternal body mass index (BMI). Notable correlations included Attractin (ATRN) with r = 0.40 (*p* = 0.004), as well as several complement system proteins: Complement C3 (C3) with r = 0.48 (*p* < 0.001), Complement factor B (CFB) with r = 0.46 (*p* < 0.001), and Complement factor I (CFI) with r = 0.51 (*p* < 0.001). Other proteins also showed significant associations with maternal BMI, including Hemopexin (HPX, r = 0.48, *p* < 0.001), Heparin cofactor 2 (SERPIND1, r = 0.43, *p* = 0.002), Pregnancy zone protein (PZP, r = 0.44, *p* = 0.002), Vitamin K-dependent protein S (PROS1, r = 0.43, *p* = 0.002), and Pigment epithelium-derived factor (SERPINF1, r = 0.61, *p* < 0.001). These findings highlight the intricate relationships between serum protein levels, patient demographics, and clinical parameters, offering insights that could help in the understanding of the biological mechanisms underlying various conditions in pregnant patients. The strong correlations, especially concerning maternal BMI, suggest that these proteins may serve as important biomarkers for further research into maternal health and its implications during pregnancy.

### 2.3. Building of a PE Prediction Model Based on the First-Trimester Maternal Serum Proteome

To improve early prediction of preeclampsia (PE), we developed a highly effective model by analyzing maternal serum proteins from the first trimester. This model utilizes a Support Vector Machine (SVM) and integrates the concentrations of 19 specific proteins, as shown in Appendix A. Our approach involved optimizing the protein set size to maximize both accuracy and the combined sensitivity and specificity. In our PE prediction model, although the three-protein and fourteen-protein sets achieved accuracy levels close to what was attained with the nineteen-protein set, the latter significantly exceeded others in terms of combined sensitivity and specificity (Appendix A). Notably, the model demonstrated an area under the curve (AUC) of 0.91, with a sensitivity of 87% and specificity of 95%, highlighting its robust diagnostic capabilities (see Table 2 for further details). Within this predictive framework, proteins such as APOM, F2, and AZGP1 were identified as significantly altered before the clinical onset of PE, underscoring their potential as early biomarkers for the condition (see Figure 1 and Appendix A for detailed statistical analysis and visual representation).

To estimate hazard ratios (HRs) and 95% confidence intervals (CIs) for the occurrence of PE, we employed the Cox proportional hazards regression model, using gestational age as the time axis. This approach allowed us to generate appropriate risk sets, with the censoring of subjects who developed PE. We assessed the statistical significance of the HRs’ deviation from one using the Wald statistics.

For the routine first-trimester preeclampsia screening test recommended by the FMF, which incorporates maternal factors, mean MAP, mean UtA-PI, serum PlGF, and serum PAPP-A, the AUC was 0.58 and the HR was 1.4 (CI: 0.7–3.0), showing no statistical significance (*p* = 0.43).

Conversely, the serum protein-based SVM model demonstrated a substantially higher HR of 13.5 (CI: 4.6–40.8), with a Wald statistics *p*-value of less than 0.001, indicating strong statistical significance (see Figure 4 and Table 2 for details).

Additionally, Kaplan–Meier curves stratified by positive and negative results of the routine FMF screening test showed no statistical significance (*p* = 0.34). However, the Kaplan–Meier curves for the serum protein-based SVM model revealed statistically significant differences, as confirmed by the log-rank test (*p* < 0.0001).

To further evaluate the timing of PE onset risk, an additional SVM model was developed. This model is designed to differentiate between early- and late-onset PE, with positive results indicating a prognosis for eo-PE and negative results indicating lo-PE. The model incorporates the concentrations of 16 specific proteins, as detailed in Appendix A.

This SVM model exhibited remarkable performance, achieving an area under the curve (AUC) value of 0.89, with 88% sensitivity and 91% specificity, providing a comprehensive measure of its diagnostic accuracy, as demonstrated in Appendix A. Furthermore, the HR calculated by this model was 25.7 (95% CI: 3.01–208.1), with a Wald test *p*-value of 0.003, indicating strong statistical significance. Additionally, the comparison of survival curves using the log-rank test, as shown in Appendix A, further confirmed the statistical significance of the model’s prognostic capabilities, with a *p*-value less than 0.001.

## 3. Discussion

The primary goal of medicine is to identify the causes of diseases and predict their occurrence before clinical symptoms arise [14]. PE is classified as one of the major obstetric syndromes, wherein a variety of pathological processes trigger specific signaling pathways that lead to the clinical presentation of the disease [15]. The manifestation of the disease can vary significantly in its biochemical profile and clinical progression, depending on the individual patient and the timing during pregnancy when symptoms appear [16]. Early-onset PE (eo-PE), which occurs before 34 weeks of gestation, is relatively rare, affecting only 0.38% of all pregnancies. In contrast, late-onset preeclampsia (lo-PE), which typically follows a less severe trajectory and usually results in better outcomes, is seven times more common [16,17]. Numerous studies have thoroughly documented the correlations between early-onset preeclampsia and factors such as chronic hypertension and nulliparity. Similarly, lo-PE has been consistently linked to obesity and chronic hypertension [16].

The most widely accepted theory of the etiology and pathogenesis of eo-PE posits that it results from inadequate placentation, which is caused by insufficient remodeling of the spiral arteries. This leads to impaired angiogenesis within the placenta, resulting in decreased uteroplacental blood flow and episodes of hypoxia/reperfusion [18,19]. Placental dysfunction causes the production and release of reactive oxygen species, cytokines, lipid peroxidases, endothelin-1, and sFlt-1 (a natural antagonist of VEGF). Ultimately, placental hypoperfusion leads to generalized endothelial dysfunction and growth restriction in the fetus [3,20].

Numerous studies aimed at identifying risk factors for PE have consistently reported a lack of clear etiological significance for these factors [21,22]. This ambiguity has significantly hindered the development of effective targeted prevention strategies, as understanding the underlying mechanisms of the disease is crucial for intervention. The identification of reliable marA systematic review conducted by Navajas et al. in 2021 presented a panel of 63 proteins whose levels are consistently altered in the context of PE [22]. This comprehensive analysis of proteins derived from blood, urine, and placenta could pave the way for the establishment of biomarkers that not only enhance diagnostic accuracy but also facilitate the monitoring of disease progression and response to treatment. Speculatively, the identification of these proteins could lead to exciting new avenues in PE management. For instance, if specific proteins are found to correlate strongly with the eo-PE, targeted therapies could be developed that modulate these biological markers, potentially preventing the disease from developing in at-risk pregnancies.

In the near future, advancements in technologies, such as machine learning and artificial intelligence, may allow for the identification of patterns within vast datasets of clinical and proteomic information. This could enhance our ability to stratify patients according to their risk for developing PE, enabling personalized monitoring and tailored prevention strategies. Ultimately, a multidimensional approach that incorporates insights from molecular biology, clinical research, and advanced data analysis will be essential for unraveling the complexities of preeclampsia and improving outcomes for both mothers and their infants.

In this study, a quantitative analysis of 125 serum proteins was performed using a commercially available BAK 125 kit. Notably, 62% (18) of these proteins are among the 29 serum/plasma proteins identified as potential PE markers in the systematic review conducted by Navajas et al. (2021) [22]. These proteins include SERPINA1, Alpha-2-antiplasmin, Alpha-2-macroglobulin, Apolipoprotein B-100, Apolipoprotein E, Clusterin, C1QB, Complement factor B, Fibrinogen gamma chain, FN1, Coagulation factor IX, Glutathione peroxidase 3, Pigment epithelium-derived factor, Plasma protease C1 inhibitor, Pregnancy zone protein, Protein AMBP, Serotransferrin, and Transthyretin. However, it is important to note that the aforementioned systematic review encompassed studies involving plasma/serum samples collected throughout all trimesters of pregnancy. In contrast, the current study exclusively utilized samples obtained between the 11th and 13th weeks of pregnancy.

Ten serum proteins have been identified as potential early indicators for PE. Remarkably, nine out of these ten proteins (90%) have shown significant changes in PE as reported by other studies using plasma/serum semi-quantitative proteomics. These proteins include C1QB [23], F2 [23,24,25], SERPINA1 [25,26,27,28], APOM [23], LYZ [27,29], ALB [28], AZGP1 [25], ATRN [30], and APOA4 [23]. The consistency of these findings across multiple studies underscores their potential utility as biomarkers for the early detection of PE.

The proposed ten PE biomarkers are integral to various interconnected molecular processes, collectively forming a cohesive molecular network, as illustrated in Figure 3C. This network underscores the interconnected nature of these biomarkers and highlights their collective contribution to the pathophysiology of early-onset PE. Understanding these relationships provides deeper insight into the molecular mechanisms underpinning PE and could inform the development of early diagnostic tools and targeted therapeutic strategies.

The molecular processes upregulated in PE prior to onset include complement and coagulation cascades; platelet activation, signaling, and aggregation; and the regulation of insulin-like growth factor transport and uptake by insulin-like growth factor binding proteins. Additionally, vitamin B12, selenium, and folate-related metabolic pathways are involved. These molecular processes are indicative of all four proposed subtypes of PE, as described by Than NG and colleagues in their 2022 study: the placental and metabolic subtype (which exhibits the most severe disease progression), the maternal anti-fetal rejection-type preeclampsia, and the extracellular matrix-related preeclampsia (primarily comprising late-onset cases) [24]. All of this reflects the heterogeneous nature of preeclampsia and suggests that the current study likely included patients from all four molecular subtypes of PE.

Insulin signaling (FDR = 0.006) is a crucial feature of the metabolic subtype of PE. Elevated first-trimester serum APOA4 levels have been detected in early PE, suggesting a link to its metabolic nature. APOA4 influences lipoprotein metabolism and cholesterol transport and exhibits anti-inflammatory properties [31,32]. Another apolipoprotein, APOM, is overexpressed before PE onset, contributing to defective placentation and endothelial dysfunction through its roles in inflammation, lipid transport, and metabolism [23,33]. APOM plays an essential role in defective deep placentation and endothelial dysfunction by controlling inflammatory response, lipid transport, and metabolism [34]. PE is marked by dyslipidemia, with elevated triglycerides and altered HDL cholesterol [35,36]. Our recent study on IUGR markers shows increased APOA4 in late-onset IUGR, highlighting lipid and glucose metabolism disruptions in both PE and IUGR [37].

Additionally, APOA4 acts as a ligand for platelet integrin αIIbβ3, exerting a negative regulatory effect on thrombosis [32]. Its interaction with aspirin could lead to improved thrombosis treatments, potentially offering new therapeutic avenues for PE [38]. The elevated APOA4 levels seen in PE and IUGR might serve a protective role for both mother and fetus in managing the prothrombotic conditions often present during pregnancy [37]. Interestingly, the platelet activation pathway (FDR = 0.004), which was notably overrepresented in this study, was the only pathway common to all four PE subtypes [24]. This observation is in accordance with previous findings that report increased platelet volumes and platelet activation in preeclampsia [39]. Thrombocytopenia in PE is thought to result from the excessive consumption of platelets due to their increased activation, which is triggered by widespread endothelial injury. In severe cases, this condition can progress to disseminated intravascular coagulation, further complicating the scenario by causing lower levels of fibrinogen and antithrombin, and increased prothrombin time and fibrin degradation products [3].

Increased platelet activation activates the trophoblastic inflammasome and leads to placental fibrinoid deposition [40]. This series of events induces systemic maternal inflammation and increases thrombin generation [41]. Supporting these findings, an elevated abundance of F2, a pro-coagulant protein, has been detected in maternal circulation from the first trimester through to gestation, which is consistent with the conclusions drawn by Than N.G. et al. in their 2022 study [24]. These results align with the fundamental pathogenesis of PE, characterized by an earlier and more pronounced activation of coagulation pathways compared to that observed in normal pregnancies [42]. In physiological pregnancies, the coagulation system is naturally modulated to maintain a balance that supports both maternal and fetal health. However, in PE, this balance is disrupted, leading to excessive platelet activation, increased thrombin generation, and widespread endothelial injury. This pathological coagulation cascade not only contributes to the clinical manifestations of PE but also exacerbates systemic inflammation and vascular complications, distinguishing PE from normal gestational processes [3].

A wide variety of proteins are involved in the modulation of these molecular pathways. Among them, C1QB merits special attention, not only as a crucial component in the activation of the classical complement pathway but also as a receptor (CD93) that is abundantly expressed in the maternal decidua, playing a significant role in regulating angiogenesis and the invasion of trophoblast cells [43]. Pregnant C1q-deficient mice exhibit characteristics of P, such as hypertension, albuminuria, decreased blood levels of pro-angiogenic VEGF, and elevated levels of the anti-angiogenic factor sFlt-1 [44]. The multidirectional changes in the levels of this protein during the first trimester of pregnancy—when comparing PE cases to a control group—identified in various studies indicate the necessity for further investigation through multicenter research efforts [45,46,47,48].

SERPINA1, a serine protease inhibitor, is actively studied in the context of preeclampsia (PE) [49,50,51,52,53,54,55,56,57,58]. A study by Kolialexi A. et al., 2017, observed elevated levels of SERPINA1 during the first trimester in patients who later developed early-onset PE compared to those with normal pregnancies [59]. Our research demonstrates that SERPINA1 effectively distinguishes between early- and late-onset PE patients. Additionally, SERPHINA1 increase is a hallmark of PE manifestation [26,27,28]. Urinary SERPINA1 peptides are suggested as a non-invasive marker for assessing PE severity [50,57,60]. In the placenta, SERPINA1 modulates local inflammation, trophoblast invasion, and spiral artery transformation by inducing endoplasmic reticulum (ER) stress. Disruption in the interaction between A1AT and ER stress at the maternal–fetal interface could lead to abnormal placental development, potentially causing inadequate trophoblast invasion and resulting in hypertensive disorders and PE [49].

Moreover, recent findings indicate that misfolded proteins, including SERPINA1, amyloid β, and albumin, accumulate in the urine, serum, and placenta of women with PE [56,58]. This process is likely influenced by factors such as ischemia, hypoxia, and chronic inflammation, which contribute to protein misfolding and increase ER stress [61,62,63]. The excess of misfolded proteins can obstruct normal trophoblast invasion, resulting in aggravated ischemia and heightened endoplasmic reticulum (ER) stress. This sets off a harmful feedback loop that perpetuates complications and further disrupts placental function, ultimately compromising maternal and fetal health.

Interestingly, the placental hormone PAPP-A, which is included in FMF first-trimester screening, has demonstrated a direct association with the acute-phase protein inter-alpha-trypsin inhibitor heavy chain H2 (ITIH2), showing a correlation coefficient of r = 0.46 (*p* < 0.001). Notably, ITIH2-4 protein family members have been previously proposed as potential diagnostic markers for PE [23,28,30,64,65,66].

In the context of maternal serum albumin, elevated levels were observed in the PE group and were significantly correlated with various clinical features of PE. Specifically, albumin levels correlated with proteinuria (r = 0.49, *p* < 0.001), thrombocytopenia (r = −0.54, *p* < 0.001), mean UtA-PI (r = 0.42, *p* = 0.003), onset of PE (r = −0.41, *p* = 0.003), and newborn mass (r = −0.37, *p* = 0.009). Moreover, ALB shows the largest effect size, as measured by Cohen’s d values (Figure 2B).

There is a significant link between maternal albumin levels and PE. Albumin, a major plasma protein produced by the liver, plays a critical role in maintaining colloid osmotic pressure and serves as a carrier for various substances including hormones, drugs, and fatty acids. Serum/plasma albumin are affected by an individual’s nutritional status and metabolic needs, with factors such as inflammation and elevated nutritional risk being associated with reduced albumin levels [67]. Several studies have shown that maternal albumin levels measured at the onset of disease tend to be lower (*p* < 0.001) in women with preeclampsia compared to those with normal pregnancies [68,69]. Moreover, a recent study by Xia Y. et al., 2024, discovered an increase in hypoalbuminemia in PE patients with poor clinical outcomes at 20 weeks of gestation [69].

Hypoalbuminemia can be attributed to several factors associated with preeclampsia, including endothelial dysfunction and increased vascular permeability, proteinuria, hepatic dysfunction, and systemic inflammation. Preeclampsia is characterized by widespread endothelial dysfunction, which increases vascular permeability. This can lead to the leakage of albumin from the circulation into the interstitial space, thereby lowering plasma albumin levels. One of the hallmark signs of preeclampsia is proteinuria, where significant amounts of protein, including albumin, are lost in the urine due to glomerular endothelial injury [70]. This contributes directly to reduced serum albumin levels. Moreover, albumin is a kidney injury biomarker. The measurement of albumin excretion in urine, as opposed to total protein excretion, has become increasingly recognized as a more specific marker for assessing glomerular injury [70,71].

PE can affect liver function, leading to decreased synthesis of albumin. Although severe hepatic impairment is less common, even mild liver dysfunction can reduce albumin production. Systemic inflammation, which is a feature of preeclampsia, can alter the hepatic production of albumin. Cytokines released during inflammatory states can downregulate albumin synthesis in the liver. Nevertheless, the HALP score calculated as [hemoglobin (g/L) × albumin (g/L) × lymphocytes (/L)]/platelets (/L) [72] has been identified as an independent indicator for preeclampsia with severe features, predicting the development of such conditions with a sensitivity of 74.5% and a specificity of 81.3% [73].

It is worth noting that all existing studies on albumin levels in PE were conducted either during the manifestation of the pathology or when the effect of the pathology on the mother’s body was already pronounced (20 weeks of gestation in the case of early PE). In our work, we studied blood serum samples at 11–13 weeks of gestation. It is known that the proteomic composition of blood undergoes significant changes throughout pregnancy. Further, longitudinal studies are needed to determine the reference intervals of albumin in normal and pathological conditions at different stages of gestation. Elevated or altered albumin levels may serve as early indicators for potential complications, providing valuable insights for the early detection and management of preeclampsia and related conditions. They also emphasize the need for comprehensive management strategies that address both the symptomatic manifestations and underlying pathophysiological changes in preeclamptic patients.

The routine first-trimester PE screening test recommended by the FMF showed no statistical significance, with an HR of 1.4 (95% CI: 0.7–3.0) and a *p*-value of 0.43. To improve the early prediction of PE, a highly effective SVM model was developed, analyzing 19 maternal serum proteins from the first trimester. This model achieved an AUC of 0.91, a sensitivity of 87%, a specificity of 95%, and an HR of 13.5 (95% CI: 4.6–40.8) with a *p*-value of less than 0.001. Furthermore, an additional SVM model was created to differentiate between the early and late onset of PE, yielding an AUC of 0.89, 88% sensitivity, 91% specificity, and an HR of 25.7 (95% CI: 3.01–208.1) with a *p*-value of less than 0.01. In conclusion, these findings demonstrate that serum protein-based SVM models possess significantly higher predictive power compared to the routine first-trimester screening test, highlighting their superior utility in the early detection and risk stratification of PE.

To ensure the broader applicability of our models across different populations and healthcare settings, careful consideration is required. Although our study has shown strong performance with specific serum proteins measured at 11–13 weeks of gestation, it is crucial to undertake external validation using independent datasets. These datasets should encompass a wide range of patient demographics and larger sample sizes to verify the model’s reliability and generalizability. Increasing the number of PE samples is also vital to distinguish between patients with varying molecular subclasses of PE, which could help elucidate the underlying pathogenic mechanisms and potentially identify new diagnostic and therapeutic targets. Moreover, the proteins analyzed should be tested in a longitudinal study to assess their predictive value over time. The inclusion of additionally known marker proteins, such as placental protein 13 (PP13), PAPP-A, endoglin (ENG), vascular endothelial growth factor (VEGF), and vascular cell adhesion protein 1 (VCAM-1), may significantly enhance the accuracy of the predictive model. By integrating these biomarkers, the model’s sensitivity and specificity could be improved, leading to more reliable and comprehensive predictions.

Despite these limitations, our study underscores the significant potential of advanced protein models and provides valuable clinical insights. Firstly, the participant groups were carefully matched based on demographic characteristics and FMF-calculated PE risk, and a comparative group (GAH) was included to enhance the robustness of our findings. Secondly, we applied a targeted MRM-based quantitation of serum proteins, using the commercially available and extensively validated BAK-125 kit. The clarity and transparency of the interpretive framework inherent in this methodology significantly ease the transition of our findings into clinical practice, providing valuable contributions for both healthcare professionals and researchers alike.

## 4. Materials and Methods

### 4.1. Study Design

A pilot prospective cohort study was conducted involving 50 pregnant women who underwent first-trimester pregnancy screening between 11 and 13 weeks of gestation and who were subsequently monitored and delivered at the I. Kulakov National Medical Research Center for Obstetrics, Gynecology, and Perinatology. Based on the outcomes, four clinical groups were formed: eo-PE (*n* = 8), lo-PE (*n* = 22), GAH (*n* = 7), and physiological pregnancy (CRL, *n* = 13). Inclusion criteria for participation in this study were as follows: maternal age between 18 and 45 years, singleton pregnancy, and a gestational age of 11–13 weeks at the time of sample collection. Exclusion criteria included multiple pregnancies, a history of organ transplantation, autoimmune diseases and malignancies, fetal chromosomal abnormalities, pregnancy achieved through assisted reproductive technology, and refusal to continue participation in this study. All women included in this study provided informed consent.

The diagnosis of IUGR was made according to the Delphi criteria and clinical recommendations of the Ministry of Health of the Russian Federation [74]. PE was defined based on the International Society for the Study of Hypertension in Pregnancy (ISSHP) guidelines, which stipulate elevated blood pressure (systolic blood pressure ≥ 140 mmHg and/or diastolic blood pressure ≥ 90 mmHg on two separate occasions, at least four hours apart) accompanied by proteinuria (>300 mg/24 h or a protein/creatinine ratio > 30 mg/mmol) developing after 20 weeks of gestation [75]. Early-onset preeclampsia was classified as preeclampsia occurring before 34 weeks of gestation, while late-onset preeclampsia referred to cases that emerged at or after 34 weeks, irrespective of the gestational week at delivery. The control group comprised women with physiological pregnancies, serving as a baseline for comparison across the different conditions studied.

Participants were enrolled during the 11–13-week gestational window as part of the first-trimester pregnancy screening test. Blood serum samples were collected using SerumZ/9 tubes (Monovette, Sarstedt, Germany) and subsequently centrifuged at 300× *g* for 20 min. The resulting supernatant was stored in cryo-tubes at a temperature of −80 °C for future analyses.

Comprehensive first-trimester screening involved several assessments, including the measurement of height, weight, and BMI, as well as blood pressure readings [76]. Additionally, the pulsation index of the left and right uterine arteries (UtA-PI) was calculated through transabdominal color Doppler ultrasound [77]. The concentrations of PIGF and PAPP-A in serum were measured utilizing the Delfia Xpress system (PerkinElmer Life and Analytical Sciences, Shelton, CT, USA).

The risk of PE was assessed using data obtained from the first-trimester screening conducted between 11 and 13 weeks of gestation. This assessment was performed using the FMF calculator, which is accessible at https://fetalmedicine.org/research/assess/preeclampsia/first-trimester (accessed on 24 September 2024). The risk calculation incorporated maternal history factors, including CAH, type I or type II diabetes, systemic lupus erythematosus, and anti-phospholipid syndrome. Additionally, maternal characteristics such as age, BMI, race, and smoking status were included. Biophysical measurements taken into account were the MAP and the mean UtA-PI. Biochemical measurements involved the concentrations of serum PAPP-A and PlGF. A high risk of developing preeclampsia was defined as a calculated risk ratio greater than 1:100.

### 4.2. Serum Preparation for Quantitative Proteomics

Serum sample analysis was conducted utilizing the BAK 125 kit (MRM Proteomics Inc., Montreal, QC, Canada). This kit comprises two synthetic peptide mixtures. The first contains 125 unlabeled “light” peptides used to create calibration curves, and the second includes 125 stable-isotope-labeled internal standard (SIS) “heavy” peptides, which are introduced into each sample to facilitate accurate quantification [78]. For sample preparation, a 10 µL aliquot of serum was processed following the manufacturer’s protocol. By adhering to the standardized protocol, the accuracy, reproducibility, and reliability of the peptide quantification are maintained throughout the analysis.

### 4.3. Quantitive Analysis of 125 Serum Proteins (LC-MRM-MS)

LC-MRM-MS analysis was conducted using the ExionLC™ ultra-high performance liquid chromatography (UHPLC) system from Thermo Fisher Scientific (Waltham, MA, USA), which was paired with the SCIEX QTRAP 6500+ triple quadrupole mass spectrometer (SCIEX, Toronto, ON, Canada). The parameters for LC-MS were optimized based on methodologies established in previous studies [37,79,80].

For the analysis, a sample injection volume of 10 µL was utilized. The separation of peptides was achieved using a UPLC Peptide BEH column (C18, 300 Å, 1.7 µm, dimensions 2.1 mm × 150 mm) obtained from Waters (Milford, MA, USA). A gradient elution method was employed, with a flow rate of 0.4 mL/min over a 53 min gradient, transitioning from 2% to 45% of mobile phase B (0.1% formic acid in acetonitrile).

Mass spectrometric measurements were conducted using the MRM acquisition mode to ensure targeted detection of the peptides of interest. The electrospray ionization (ESI) source was finely tuned with the following settings: an ion spray voltage of 4000 V, a temperature of 450 °C, and an ion source gas flow rate set at 40 L/min. A comprehensive transition list for the MRM experiments—which includes retention times and the Q1/Q3 masses for each peptide—can be found in Appendix A.

The analytical batch comprised a robust structure that included not only the serum samples but also blanks, three distinct levels of quality control (QC-A, QC-B, QC-C), and eight calibration levels (designated A-H). This design ensures the reliability and validity of the results, allowing for accurate quantitative analysis of the target peptides.

LC-MRM-MS was performed using an ExionLC™ UHPLC system (Thermo Fisher Scientific, Waltham, MA, USA) coupled with a triple quadrupole mass spectrometer SCIEX QTRAP 6500+ (SCIEX, Toronto, ON, Canada). LC-MS parameters were optimized in the previous studies [37,79,80].

Skyline Quantitative Analysis (version 20.2.0.343), developed by the University of Washington, was employed for the quantitative analysis of LC-MRM-MS data [81]. The protein concentrations in the experimental samples were determined in femtomoles per microliter (fmol/µL) using a calibration curve that was established through 1/x^2^-weighted linear regression.

The chromatographic peaks corresponding to the NAT (native) and SIS (synthetic internal standard) peptides within the samples were manually evaluated. This assessment focused on verifying the shape and ensuring accurate integration of the peaks for reliable quantification. Furthermore, all MRM assays included in the BAK 125 kit were conducted in accordance with the standards established by the Clinical Proteomic Tumor Analysis Consortium (CPTAC) [82].

For additional clarity, Appendix A illustrates example MRM data alongside the corresponding calibration curves, providing a visual representation of the results and demonstrating the robustness of the analysis.

### 4.4. Data Statistical Processing

Statistical analysis and data visualization were conducted using custom R scripts (version 4.3.2). These scripts leveraged multiple packages, including xgboost, e1071, pROC, reshape2, ggplot2, corrplot, survminer, survival, and qgraph.

Proteins quantified below the LLOQ in more than 50% of the samples were excluded from the analysis. This curation step reduced the number of proteins in this study to 69 features (outlined in Appendix A). For any remaining missing values, imputation was performed by replacing them with half the LLOQ value.

For continuous clinical parameters, median values along with the first and third quartiles were calculated to summarize the data distribution. Comparisons of continuous parameters between groups were executed using the Mann–Whitney U test, while categorical parameters were compared using the Pearson chi-squared test.

The statistical significance of protein alterations was evaluated between samples from patients without PE—comprising the control and GAH groups—and samples from patients with PE, which included eo-PE and lo-PE groups. Proteins showing statistically significant differences between these groups were incorporated into pathway enrichment analysis using consensusPathDB [83].

Correlations between protein concentrations and clinical parameters were examined using the Spearman rank correlation test, applying a significance threshold of *p* < 0.05.

Protein–protein interactions were analyzed via the STRING database (accessed on 6 January 2023). Networks were constructed to include only associations with *p* < 0.05. Protein categorical annotations were sourced from GeneOntology via the SwissProt database.

### 4.5. SVM Model Development

For PE prognosis and timing of onset (before or after 34 weeks), an SVM model was employed due to its established efficacy in creating models with high diagnostic accuracy in proteomics [84,85,86]. The optimal set of features for the SVM model was determined using the recursive feature elimination method [87]. This iterative process involved the step-by-step elimination of features with the smallest absolute value of weight, rebuilding the model at each step and evaluating it through leave-one-out cross-validation. The model with the best performance metrics was chosen as the final model.

The quality of PE prediction was assessed using a Cox proportional hazards regression model and the Kaplan–Meier method [87,88]. Log-rank significance *p*-values were calculated to validate these models. HRs of PE were computed using the Cox proportional hazards regression. Additionally, Kaplan–Meier survival curves were generated for groups stratified by the model’s PE prediction (positive or negative), allowing for the comparison of pregnancy outcomes at specific time points (weeks of gestation) [16,89].

## 5. Conclusions

Utilizing the BAK-125 kit, we have successfully identified ten serum proteins, nine of which have been previously associated with PE in prior research, underscoring the consistency and reliability of our findings. These proteins play key roles in critical molecular processes, including complement and coagulation cascades, platelet activation, and insulin-like growth factor pathways, which correlate with the four proposed subtypes of PE. We observed that elevated maternal albumin levels at 11–13 weeks of gestation were associated with PE features, highlighting the potential of albumin as an early biomarker. Our SVM models exhibited high prognostic accuracy, further validating the efficacy of serum-based proteomic profiling for early PE detection, surpassing traditional screening methods. This work not only lays the groundwork for developing more effective diagnostic tools but also opens avenues for innovative therapeutic strategies. Ultimately, our findings aim to significantly improve maternal and fetal health outcomes for those affected by PE.

## Figures and Tables

**Figure 1 ijms-25-10653-f001:**
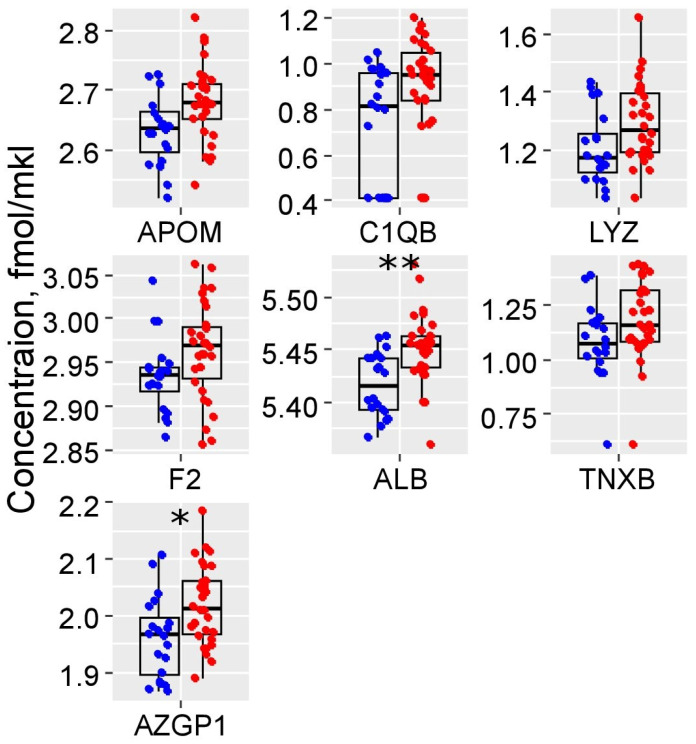
Box plots of significantly elevated (*p* < 0.05) proteins in the first-trimester maternal blood serum in PE cases: blue dots—control and GAH group (*n* = 20), red dots—PE group (*n* = 30). *—*p* < 0.01, **—*p* < 0.001. APOM—Apolipoprotein M, C1QB—Complement C1q subcomponent subunit B, LYZ—Lysozyme C, F2—Prothrombin, ALB—Serum albumin, AZGP1—Zinc-alpha-2-glycoprotein.

**Figure 2 ijms-25-10653-f002:**
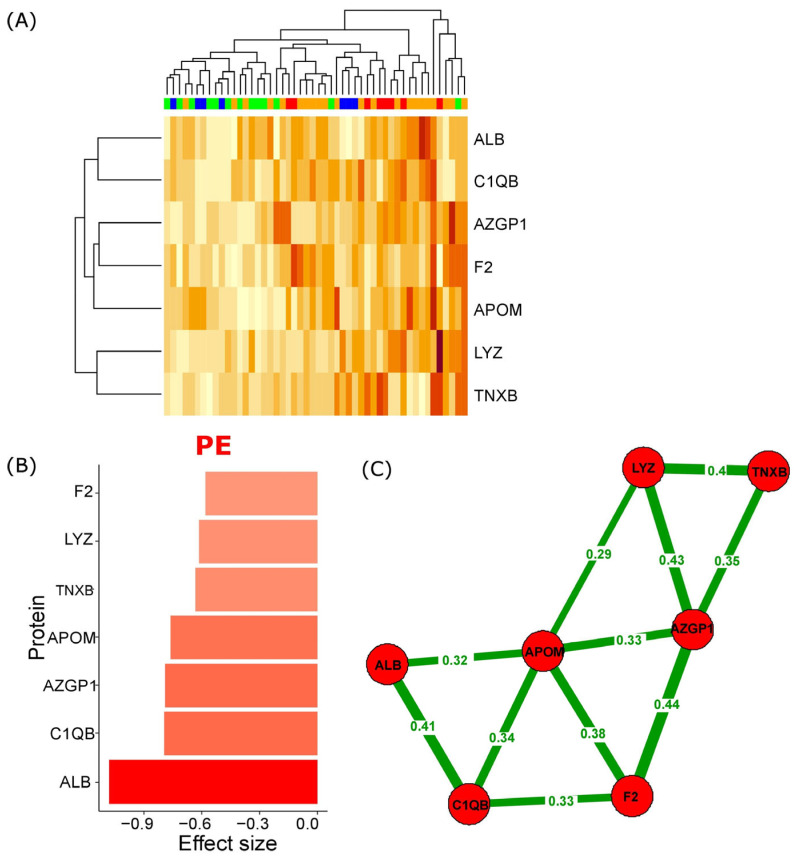
Early gestational serum protein alterations as predictive markers for PE development. (**A**) Heatmap of protein concentration distribution across samples in Euclidian distance with Ward’s minimum variance method clusterization. Green—CTR, blue—GAH, orange—lo-PE, red—eo-PE groups. (**B**) The effect sizes (Cohen’s d) among the seven proteins showed significant variation between the PE and non-PE groups. (**C**) Statistically significant associations between these proteins (Pearson test).

**Figure 3 ijms-25-10653-f003:**
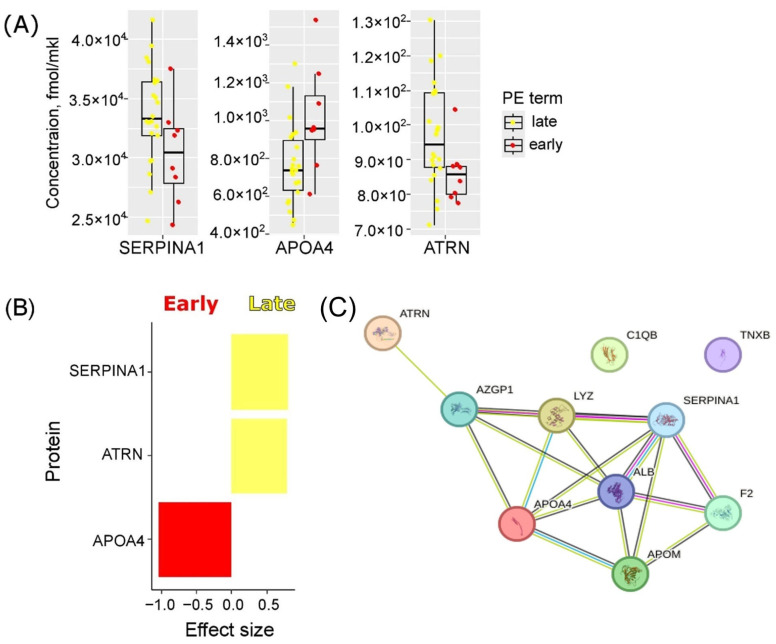
Maternal serum proteome changes associated with PE clinical types. (**A**) Boxplots of serum proteins, dramatically disturbed in early vs. late PE (*p* < 0.05). Yellow color corresponds to the lo-PE group (*n* = 22), red color—eo-PE (*n* = 8). The median is presented as a horizontal line in the interquartile range box with standard deviation whiskers. (**B**) The effect sizes (Cohen’s d) among the proteins showed significant variation between lo-PE and eo-PE groups. (**C**) Protein–protein interaction network for 10 differentially expressed plasma proteins in IUGR obtained using STRING database 11.00 (https://string-db.org/, accessed on 6 January 2023). Nodes represent proteins and edges represent protein–protein associations: purple indicates experimentally determined interactions, blue indicates interactions from the curated database, black indicates the co-expression of genes, and green indicates text mining—the result of parsing full-text articles from the PMC Open Access Subset, PubMed abstracts, summary texts from OMIM (OMIM.org), and SGD (Saccharomyces Genome Database) entry descriptions.

**Figure 4 ijms-25-10653-f004:**
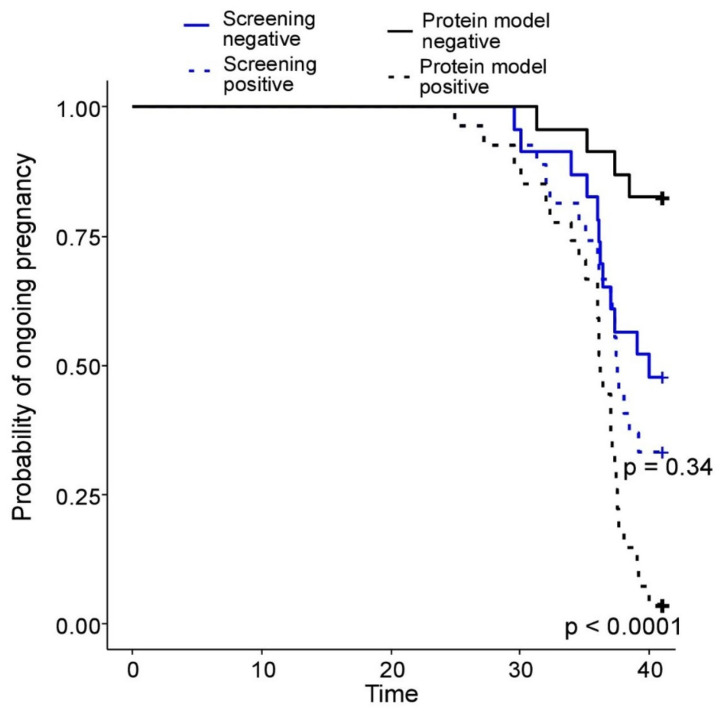
Comparison of the first-trimester PE prognosis methods: routine FMF screening (blue), SVM model based on the level of 19 maternal serum proteins (black): Kaplan–Meyer survival curves for groups with positive/negative results of each prognostic method. *p*-value was calculated according to the log-rank test.

**Table 1 ijms-25-10653-t001:** Characteristics of the study groups. eo-PE—early PE, lo-PE—late PE, GAH—gestational arterial hypertension, BMI—body mass index, HAG—chronic arterial hypertension, SBP—systolic blood pressure, DBP—diastolic blood pressure, ALT—alanine aminotransferase, AST—aspartate aminotransferase, LDH—lactate dehydrogenase, sFlt-1—soluble fms-like tyrosine kinase-1, PlGF—placental growth factor, IUGR—intrauterine growth restriction.

Feature	Group 2, eo-PE (*n* = 8)	Group 2, lo-PE (*n* = 22)	Group 3, GAH (*n* = 7)	Group 4, CTR (*n* = 13)	*p*-Value
Age, years, Me [Q1; Q3]	36.0 [34.2; 38.3]	33.5 [29.0; 37.0]	37.0 [36.5; 37.5]	34.0 [32.0; 36.0]	>0.05
BMI, Me [Q1; Q3]	25.5 [23.5; 26.25]	26.0 [23.0; 29.0]	29.0 [29.0; 33.5]	28.0 [25.0; 30.0]	*p*_13_ = 0.006
Previous PE/IUGR *n* (%)	2 (25.0%)	2 (19%)	1 (14.3%)	0	>0.05
HAG, Me [Q1; Q3]	2 (25.0%)	5 (22.7%)	0	0	>0.05
Nulliparous, *n* (%)	6 (75.0%)	15 (68.2%)	4 (57.2%)	8 (61.5%)	>0.05
High rick of PE (first-trimester prenatal screening), *n* (%)	5 (62.5%)	13 (59.0%)	3 (42.9%)	6 (46.2%)	>0.05
Max. SBP, Me [Q1; Q3]	160 [150; 165]	150 [140; 160]	145 [143; 150]	120 [115; 127]	>0.05
Max. DBP, Me [Q1; Q3]	110 [100; 110]	95 [90; 100]	90 [90; 95]	80 [75; 85]	*p*_12_ = 0.04 *p*_14_ < 0.001
Proteinuria, g/L, Me [Q1; Q3]	2.2 [1.3; 2.8]	1.2 [0.4; 2.3]	0 [0; 0.05]	0 [0; 0]	*p*_13_ = 0.001 *p*_14_ < 0.001
Platelet count, Me [Q1; Q3]	196 [153; 224]	216 [146; 253]	270 [239; 307]	247 [226; 286]	*p*_13_ = 0.04
ALT, Me [Q1; Q3]	38.8 [14.5; 66.5]	16.3 [12.5; 21.1]	16.2 [13.4; 18.5]	24.0 [14.6; 27.4]	>0.05
AST, Me [Q1; Q3]	27.4 [21.0; 54.6]	21.4 [15.7; 24.7]	14.9 [14.2; 18.6]	17.3 [7.8; 19.7]	*p*_13_ = 0.01 *p*_14_ = 0.01
LDH, Me [Q1; Q3]	452 [366; 565]	424 [372; 445]	368 [342; 397]	269 [134; 347]	*p*_14_ < 0.001
sFlt-1/PlGF, Me [Q1; Q3]	423.9 [342.3; 525.5]	128.7 [100.4; 213.0]	35.8 [27.9; 50.8]	28.5 [21.8; 45.0]	*p*_12_ = 0.003 *p*_13_ = 0.001 *p*_14_ < 0.001
HELLP syndrome, Me [Q1; Q3]	1 (12.5%)	0 (0.0%)	0 (0.0%)	0 (0.0%)	>0.05
IUGR, Me [Q1; Q3]	7 (87.5%)	4 (18.2%)	0 (0.0%)	0 (0.0%)	*p*_12_ = 0.002 *p*_13_ = 0.004 *p*_14_ < 0.001
Premature birth, *n* (%)	8 (100%)	6 (27.3%)	0 (0.0%)	0 (0.0%)	*p*_12_ = 0.002 *p*_13_ < 0.001 *p*_14_ < 0.001
Gestational age at birth, wks, Me [Q1; Q3]	31.3 [30.2; 32.2]	37.5 [36.7; 38.3]	39.0 [38.7; 39;3]	38.4 [38.2; 39.0]	*p*_12_ < 0.001 *p*_13_ < 0.001 *p*_14_ < 0.001
Emergency cesarean section, *n* (%)	8 (100%)	11 (50.0%)	0 (0.0%)	2 (15.3%)	*p*_12_ = 0.04 *p*_13_ < 0.001 *p*_14_ < 0.001
Newborn mass, g, Me [Q1; Q3]	1215 [1123; 1385]	2937 [2565; 3224]	3410 [3251; 3550]	3290 [3042; 3612]	*p*_12_ < 0.001 *p*_13_ < 0.001 *p*_14_ < 0.001
Apgar, 5 min, scores, value: *n* (%)	8:4 (50.0%) 7:4 (50.0%)	9:16 (72.7%) 8:6 (17.3%)	9:7 (100%)	9:13 (100%)	*p*_12_ < 0.001 *p*_13_ = 0.009 *p*_14_ < 0.001

**Table 2 ijms-25-10653-t002:** Prognostic performance of the first-trimester models (routine FMF screening, 19 proteins-based SVM model) across clinical groups.

Model	Predicted Outcome	Clinical Group			
		Control (*n* = 13)	GAH (*n* = 7)	Late PE (*n* = 22)	Early PE (*n* = 8)
Routine screening	not PE	7 (54%)	4 (57%)	9 (41%)	3 (38%)
	PE	6 (46%)	3 (43%)	13 (59%)	5 (62%)
SVM model	not PE	12 (92%)	7 (100%)	3 (14%)	1 (13%)
	PE	1 (8%)	0 (0%)	19 (86%)	7 (87%)

## Data Availability

Data are contained within the Appendix A.

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
