# Peer review of "First-Trimester Preeclampsia-Induced Disturbance in Maternal Blood Serum Proteome: A Pilot Study"

_ijms, 2024, doi:10.3390/ijms251910653_

Round 1

Reviewer 1 Report

Comments and Suggestions for Authors

The submitted paper explores the proteomic profile of 37 women with different phenotypes of PE and 13 healthy controls. The paper is of great interest and I commend thee Authors for their nice work. The major flaw of this work is the discussion of the findings.

In the discussion section Authors used the first part as an extension of the introduction section (with definition of PE, clinical manifestations etc.. from line 323 to 350). I would delete this part since it does not belong to the discussion.   

Furthermore, several paragraphs are dedicated to generic explanations (such as low platelets, dysregulation of complement etc.). These explanations are too broad and should be removed in favor of specific pathways under study. Authors should focus on the 10 most significant proteins under study. E.g. C1QB should be specifically discussed, and not just the complement system (several studies have evaluated the role of C1q and its receptors, such as the recent CD93, whose levels are significantly different in PE, see doi: 10.1111/aji.12586. doi: 10.1007/s40292-023-00608-y.). The same goes for APOA4, APOM, LYZ, TNXB etc...   

Author Response

The submitted paper explores the proteomic profile of 37 women with different phenotypes of PE and 13 healthy controls. The paper is of great interest and I commend thee Authors for their nice work. The major flaw of this work is the discussion of the findings.

  1. In the discussion section Authors used the first part as an extension of the introduction section (with definition of PE, clinical manifestations etc.. from line 323 to 350). I would delete this part since it does not belong to the discussion.

Answer. Thank you for reviewing our article and providing valuable comments. We have deleted this part of Discussion section. The Introduction and Discussion sections were reorganized accordingly.

  1. Furthermore, several paragraphs are dedicated to generic explanations (such as low platelets, dysregulation of complement etc.). These explanations are too broad and should be removed in favor of specific pathways under study. Authors should focus on the 10 most significant proteins under study. E.g. C1QB should be specifically discussed, and not just the complement system (several studies have evaluated the role of C1q and its receptors, such as the recent CD93, whose levels are significantly different in PE, see doi: 10.1111/aji.12586. doi: 10.1007/s40292-023-00608-y.). The same goes for APOA4, APOM, LYZ, TNXB etc...

Answer. The Discussion section was rewritten to include the description and comparison with other studies for most of 10 proteins, proposed as potential biomarkers of PE.

Reviewer 2 Report

Comments and Suggestions for Authors

This is a very well written paper that addresses an interesting and important research field (biomarkers of preeclampsia and related outcomes). No major weaknesses are identified; two small improvements could be made. One would be to shorten the introduction considerably and move the text regarding other efforts for biomarker discovery for this patient setting to the discussion section, and adding some comparisons of this novel method vs others. A second would be to describe whether there is a necessity to include all 10 of the identified proteins in the prediction models, or is only a subset could achieve the same ROC curves or close to it. The more parsimonious the protein profile is the more a faster turnaround or lower cost approach could be applied. Otherwise, its a very nice paper that has been thoughtfully developed and presented.

Author Response

This is a very well written paper that addresses an interesting and important research field (biomarkers of preeclampsia and related outcomes). No major weaknesses are identified; two small improvements could be made.

  1. One would be to shorten the introduction considerably and move the text regarding other efforts for biomarker discovery for this patient setting to the discussion section, and adding some comparisons of this novel method vs others.

Answer. The Introduction and Discussion sections were reorganized in accordance to your comment.

  1. A second would be to describe whether there is a necessity to include all 10 of the identified proteins in the prediction models, or is only a subset could achieve the same ROC curves or close to it. The more parsimonious the protein profile is the more a faster turnaround or lower cost approach could be applied. Otherwise, its a very nice paper that has been thoughtfully developed and presented.

Answer. Thank you for your insightful feedback on our study. We developed a Support Vector Machine (SVM)-based model for predicting preeclampsia, focusing on optimizing the protein set size. We aimed to maximize both accuracy and the combined sensitivity and specificity. In our PE prediction model, while the 3-protein and 14-protein sets achieved accuracy levels close to the maximum attained with the 19-protein set, the 19-protein set significantly excelled in terms of combined sensitivity and specificity (Figure S3a). This information has been included in the results section.

Round 2

Reviewer 1 Report

Comments and Suggestions for Authors

Authors have appropriately addressed my concerns and the manuscript has significantly improved.